# Analysis on Risk Characteristics of Traffic Accidents in Small-Spacing Expressway Interchange

**DOI:** 10.3390/ijerph19169938

**Published:** 2022-08-12

**Authors:** Yanpeng Wang, Jin Xu, Xingliang Liu, Zhanji Zheng, Heshan Zhang, Chengyu Wang

**Affiliations:** 1School of Traffic and Transportation, Chongqing Jiaotong University, Chongqing 400074, China; 2Chongqing Key Laboratory of “Human-Vehicle-Road” Cooperation and Safety for Mountain Complex Environment, Chongqing Jiaotong University, Chongqing 400074, China

**Keywords:** traffic engineering, traffic safety, expressway, small-spacing interchange, cause analysis of traffic accidents

## Abstract

Many small-spacing interchanges (SSI) appear when the density of the expressway interchanges increases. However, the characteristics of traffic accidents in SSI have not been explained clearly. Therefore, this paper systematically takes the G3001 expressway in Xi’an as the research object to explore the accident characteristics of SSI. Firstly, the expressway is divided into four sections. Furthermore, their safety can be evaluated by the number of accidents per unit distance of 100 million vehicles (*NAP*). Subsequently, eight indexes, such as mean spacing distance (MSD), are selected to explain the cause affecting expressway safety by developing the least square support vector machine (LSSVM). Secondly, the difference between SSI and normal-spacing interchanges (NSI) is clarified by statistical analysis. Finally, LSSVM, random forest, and logistic regression models are built using 12 indicators, such as the time spent exploring the causes of serious accidents. The results show that the inner ring *NAP* in Sections I and II with SSI is 27.2 and 33.7, higher than in other sections. The density, annual average daily traffic, and MSD adversely affect expressway traffic safety. The road condition mainly influences the serious traffic accidents in the SSI. This study can provide the theoretical basis for traffic management and accident prevention in the SSI of the expressway.

## 1. Introduction

The small-spacing interchange (SSI) is where the minimum spacing distance between adjacent interchanges is less than 1 km [1]. In recent years, many expressways have been built in Chinese cities such as Chengdu, Chongqing, and Xi’an to meet the cities’ rapid development. With the length of the expressway increasing, the expressway network around the town has become denser and denser. At the same time, the spacing distance between the interchange has to be shortened due to the terrain, administrative division, and travel demand. Thus, the small-spacing interchange has become an inevitable social demand. Yet, the safety of the SSI is not satisfactory. The paper [2] has shown that the SSI is where many accidents happen. Therefore, it is necessary to carry out a systematic accident analysis of the SSI.

At present, expressway interchange accidents have been studied comprehensively and deeply. Many countries started early in the study of traffic accidents on expressway interchanges. The cause and severity of accidents analysis are developed based on traffic accident data to explore the relationship among factors such as people, vehicles, roads, environment, and their combination. According to relevant studies and statistics, road factors account for 37–41% of the overall influencing factors. Therefore, road factors are the critical research object for influencing factors of expressway traffic accidents [3]. Meanwhile, extensive attempts have been made to study the severity and cause of the accident by constructing models, including statistical methods [4,5] and machine learning methods [6,7]. Unfortunately, the SSI is a particular interchange appearing in recent years, and the study on the characteristics of accidents has not been carried out thoroughly. Many researchers have focused on traffic characteristics’ impact on the risk in the interchange short weaving area [1]. However, the accident characteristics in SSI have not been systematically explained, such as the number of accidents, accident type characteristics, accident cause, and accident severity characteristics.

Therefore, this paper systematically investigates the characteristics of traffic accidents in the SSI. The traffic and accident data were collected on the Xi’an G3001 expressway from 2015 to 2018. Some new indexes are adopted to describe the safety of the expressway. Furthermore, through statistics, the characteristics of accidents in the SSI are explored from the aspects of accident types, spatial location distribution, etc. In addition, LSSVM, random forest (RF), and logistic regression (LR) models were developed to study the importance of factors related to accident severity in the SSI. This paper will further explain the causes and characteristics of SSI accidents and distinguish the differences between SSI and NSI accidents. Additionally, the causes of serious accidents in SSI will be studied by developing models simultaneously. 

## 2. Literature Review

Although traffic accidents are characterized by contingency, rarity, and randomness, the cumulative effect of many traffic accidents has statistical significance. Therefore, many researchers have used accident statistics to study expressway safety. Claros et al. [8,9] made a comparative analysis of the accident data before and after the reconstruction of the rhomboid interchange. They found that the number and severity of accidents in the split rhomboid interchange were significantly lower than in the traditional rhomboid interchange. Sadia et al. [10] defined the interchange complexity index (ICI) to evaluate the complexity of the interchange and explain the relationship between the number of accidents and the ICI. Wang et al. [11] used a multi-level Poisson lognormal distribution accident model to predict real-time accident risk. The study showed that average turning angle, traffic volume, downhill slope, and wet road surface would increase the accident risk level of the ramp. Qi [12] utilized interactive highway safety design model (IHSDM) software containing statistical models to evaluate interchange safety according to the accident data of expressway interchanges. The accident statistics method is mainly based on actual traffic accident data, and this method can reveal the law and characteristics of traffic accidents. Thus, the results can positively affect the safety management and design of interchange.

To carry out accident prevention, some researchers attempt to construct logistic regression (LR) and other models based on accident data to analyze accident causes. Casado-Sanz et al. [13] investigated and analyzed 1064 traffic accidents with the highest death rate. Using cluster analysis and multiple Logistic models, they found that the low traffic volume, the high proportion of large vehicles, and wide lanes would increase driver injury severity. Xu et al. [14] use the random parameter logistic regression method to establish a real-time collision risk model that integrates various collision mechanisms under different traffic conditions. The model significantly improves both the goodness of fit and collision classification accuracy. Pljakic M et al. [15] conducted macro research on the influencing factors of traffic accidents in Novi Bad for three years. The study showed that the motor vehicle mileage, parking spaces, and the number of signalized intersections might positively affect traffic accidents. Yasmin et al. [16] take three expressways as the research objects and establish multiple logit models to test the real-time collision risk components. The model employs different real-time traffic attributes (volume, speed, lane occupancy, and environmental conditions), further improving the model’s performance. Paolo I et al. [17] analyzed accident data of two roads in Norway based on LR models. They found that drivers familiar with the route are more likely to have accidents when the following conditions are met: Heavy traffic in autumn and winter, minor intersections, and the speed limit being less than 80 km/h.

With the in-depth analysis of accident causes, researchers began to explore the cause of the accidents’ severity. According to the traffic accident-related statistics in Hawaii, Kim et al. [18] used a Log-linear model and the LR model to analyze the relationship between rollover, frontal collision, and other accident forms and the severity in 1990. Mercier et al. [19] used the LR model to analyze the relationship between age and gender on the severity of traffic accidents in the form of frontal collisions on rural roads. They found that the age of drivers, road protection facilities, and vehicle location significantly correlated with the severity of traffic accidents. Yau et al. [20] found that the accident time and road types were associated with a significant traffic accident severity for male drivers using the LR model. Ali [21] analyzed the severity of traffic accidents by using the LR model. The results show that the location significantly influences the traffic accident’s severity. With the continuous progress of machine learning technology, many methods based on machine learning have been proposed. Chang and Chen [22] established the empirical relationship between traffic accidents and geometric variables of expressways by developing the classification and regression trees (CART) model and the negative binomial regression model. Tian et al. [23,24] determined the influence of traffic volume and other factors on accident severity by constructing the support vector machine (SVM), RF model, and Bayesian network. Lin et al. [7] proposed a method based on the Frequent Pattern tree (FP-tree) to select variable features that are more likely to lead to traffic accidents. Moral-Garcia et al. [25] used the information root node variation method to obtain rule sets based on the decision tree. They found that pedestrian collision was the accident type that caused the most deaths. Furthermore, speeding was the leading cause of serious injuries.

In summary, many achievements have been made in studying expressway accident analysis. Research scenarios include interchanges, ramps, and primary sections. Statistical and machine learning methods are used to analyze the characteristics of accidents and explore the causes of accidents, making positive contributions to expressway traffic safety management. Unfortunately, the study of traffic accidents in SSI has not been entirely carried out. Currently, some researchers have begun to pay attention to SSI safety. The security of SSI was explored by analyzing the spatial distribution of the risk in the segment and the risk characteristics of the short weaving area [2]. Yet, the SSI accidents lack systematic analysis, and their features and causation are not explained clearly. To address the gap, this paper aims to clarify the accident characteristics and cause of the accident in the SSI to provide a scientific basis for accident prevention and traffic safety management.

## 3. Study Area

This study selected the Chinese G3001 expressway around Xi’an city as the research object, which steps over several districts, as depicted in Figure 1. Traffic and crash data on G3001 from January 2015 to January 2018 were collected. During this time, traffic data were given in 1 h, including vehicle count, vehicle type, and average velocity. A total of 1597 crash cases were obtained to study accidents.

There are some differences in traffic operation and road infrastructure in different administrative areas of the expressway. Thus, it is necessary to divide G3001 into multiple subsections. Two principles must be followed when the expressway is divided into subsections. The first principle is to divide the SSI into the same segment as much as possible. The second principle is to facilitate the collection of traffic flow data. This paper divides expressways into four sections according to detectors’ location and interchanges’ distance, as shown in Figure 1. The distribution of interchanges within each area is shown in Table 1. In Section I, there are five interchanges in total. The spacing distance between the No. 3 and 4 interchange is lower than the standard recommended value (1 km). So, they are seen as an SSI area. Similarly, the No. 7 and No. 8 interchange are SSI areas, as shown in Figure 2.

## 4. Methods

In this paper, the LSSVM model was firstly used to describe the overall safety of the expressway to analyze the significance of SSI accidents. Then, through the statistical analysis of traffic accident data, a comparative analysis was carried out from the distribution of accident mileage, the type of accident, the number of vehicles involved, and the accident’s severity. Finally, by building models, the causes of serious accidents were analyzed. This paper selected LSSVM, RF, and LR for severity accident casual analysis to make the results more convincing. For severity, there were only two categories considered, namely serious and non-serious. When building the model, the appropriate modeling method should be selected considering the intercorrelation problem, the volume of available data, calculation complexity, and classification accuracy [26]. As mentioned above, lower traffic data volume can be applied to build the classification model. LSSVM is adopted to construct the model to address the intercorrelation problem and consider the available data characteristics. In the present work, RF was applied to specify the relationship mentioned above, as it is commonly used to rank each variable’s importance [27]. It also has a better anti-overfitting ability and operational stability than traditional methods. The LR has been proven to be very stable in previous studies on traffic crashes. Many researchers have used this model because of its low data requirements, simple structure, and easy implementation and application in practice [23]. Therefore, this paper chose the above three models for analysis.

(i)LSSVM

A new support vector machine method called LSSVM was proposed to solve the problem of pattern classification and function estimation [27]. It can overcome small-sample nonlinear issues well. The way uses the least square linear system as the loss function instead of the quadratic programming method used by the traditional support vector machine. The LSSVM simplifies the computational complexity. In addition, because LSSVM uses the least square method, it is significantly faster than other versions of support vector machines. The calculation process of LSSVM is divided into four steps: (1) Determine the input and output variables, separate the training set and test set, and normalize the training data; (2) determine the kernel function and solve the equation set; (3) calculation parameters; (4) give the results. A detailed introduction of this method is presented in Appendix B.

(ii)Random forest

RF is gaining more and more attention because it is more accurate and robust to noise than single classifiers [23,28]. It consists of any number of simple trees, where the final prediction class of the test object is the prediction pattern of all the individual trees. The RF is a collection of classification trees in which each tree votes for the class with the most frequent allocation of input data. It adds an extra layer of randomness to Bagging. In addition to using different data bootstrap samples to construct each tree, random forests also change how classification trees are built. Each node uses the best partition between all variables in a standard tree. Simultaneously, it uses the best segmentation from a subset of predictors randomly selected at that node. This counterintuitive strategy performs very well compared to many other classifiers and is robust to overfitting. Another advantage of RF is its ease of use, as it has only two parameters: the number of variables in a random subset of each node and the number of trees in the forest. The RF is not very sensitive to the values of these two parameters. A detailed explanation of it can be seen in Appendix B.

(iii)Logistic model

The logistic regression (LR) model [29] proved to be very stable in previous traffic accident risk prediction studies. The model has low data demand and a simple structure in practice. The dependent variable is the severity of traffic accidents, so the binomial Logit model is adopted. The model can be expressed as:(1)$$y=\frac{1}{1+e^{-(\alpha +\beta X)}}$$
where y is the predicted probability belonging to the default class. $1/1+e^{-(\alpha +\beta X)}$ is an S-shaped function. It takes any real value as a parameter and maps it to a range between 0 and 1. α+βX is a linear model in logistic regression. α is the intercept term, X is the independent variable vector, and β is the corresponding coefficient vector. In addition, the decision boundary is a threshold used to classify the probabilities of logistic regression into discrete classes. The decision boundaries are as follows:(2)$$y=\left\{ \begin{array}{l} 0,\;if\;probability\;<\;0.5\\ 1,\;if\;probability\;>0.5 \end{array}\right.$$

## 5. Result

Previous studies [1,26] have not confirmed whether the cause of the accident in the SSI area is consistent with that in the NSI area. So, this paper will clarify the difference by analyzing the accident data.

### 5.1. Significance Analysis of SSI Accidents

Traffic volume and section length may affect the number of traffic accidents, so it is necessary to consider the influence of the above two factors when describing the safety of expressways. Therefore, the accident rate of 100 million vehicles per kilometer (*NAP*) is selected as the safety evaluation index in this paper to evaluate safety in the four sections quantitatively. The specific calculation formula is as follows:(3)$$NAP=\frac{N}{q\times L}\times 10^{8}$$
where *N* represents the number of traffic accidents, *q* stands for traffic volume, and *L* is road length. 

The G3001 expressway has two directions: the inner ring (IC) and the outer ring (OC). The data samples are collected in two directions, and there are specific differences in traffic flow data in different directions. However, there is no significant difference in road geometry and traffic environment between inner and outer circles. The result of *NAP* is presented in Figure 3.

Figure 3 shows that the top three *NAP* average values were II-IC, I-IC, and III-IC. From 2015 to 2018, the *NAP* value of I-IC increased rapidly in four consecutive periods, reaching 51.9 in 2018 and higher than in other regions. The *NAP* value of II-IC was higher than that in other regions for four successive years. Sections I and II both have SSI. Therefore, it is necessary to analyze the cause of the accident to clarify the impact of the interchanges’ spacing distance on the *NAP* value.

Based on existing studies and collected data, this paper selects eight factors for analysis, as shown in Table 2. In Sections I and II, the spacing distance of multiple interchanges is lower than the standard specification value, and the value of *NAP* is relatively high. The spacing distance of interchanges is likely to affect the number of accidents, so the mean spacing distance (MSD) of interchanges is selected as one of the influencing factors. In addition, AADT, velocity (VEL), density (DEN), interweaving ratio (IR), section length (SL), the proportion of non-sunny weather (NSW), and the number of accidents (NOA) were selected as variables to analyze the cause of accidents further. AADT, VEL, DEN, IR, MSD, SL, NSW, and NOA were the independent variables, and *NAP* was the dependent variable. The eight variables are macro-level indicators, which take annual as a statistical cycle to describe the overall situation of the G3001 expressway. Subsequently, the LSSVM is developed to explore the importance of different factors, and the result is shown in Figure 4.

In Figure 4, AADT and DEN are the top two factors and have a particular impact on traffic safety, similar to previous studies [30,31]. When the density and traffic volume increase, the degree of mutual interference between vehicles increases, slowing the speed, and the traffic conflict between vehicles becomes more serious. These produced accident hidden danger and increased the probability of traffic accidents. Simultaneously, MSD is also one of the factors affecting accidents, ranking third in importance. This result verifies the abovementioned hypothesis, reflecting that MSD may impact traffic safety. When the spacing distance between the interchange shorten, the entering, exiting, and straight cars frequently interweave in the narrow space, resulting in increased conflicts between vehicles. In addition, it may positively affect drivers’ cognitive difficulties. Therefore, the characteristics of traffic accidents in the SSI and NSI may differ. The relevant influencing factors deserve further analysis and research.

### 5.2. Statistical Analysis of Accidents in SSI Area

Through the statistics of accident data in the SSI of Sections I and II, respectively, the analysis of spatial distribution from the accident location, severity, type, and the accident vehicle are summarized to find out the commonness and difference in accident characteristics of the two kinds of interchange areas. 

The statistic range of the interchange includes the interchange structure and the connection part of two interchanges. The center of the No.4 interchange is the starting position (pile No.0), and statistics are made according to the mileage position of the accident, as shown in Figure 5.

There were 271 accidents in total, as shown in Figure 5a. Furthermore, most of the accidents in Section I occurred in the interchange area. There were 109 traffic accidents in the No.5 interchange, which is the highest among the eight interchanges. There were 72 traffic accidents in the SSI, including the No.3 and No.4 interchanges, accounting for 37% of Section I accidents. On the other hand, there are three interchanges in Section II, namely, No.6, No.7, and No.8 interchanges. No.7 and No.8 interchanges are SSI. According to Figure 5b, 209 traffic accidents occurred in the interchange area and 149 accidents in the SSI, accounting for about 71%. In general, 731 accidents occurred in Sections I and II, and 452 occurred in the interchange area, accounting for 61.8%. This result again shows that the expressway interchanges are the most frequent site of accidents [32]. Meanwhile, 193 traffic accidents occurred in the SSI, accounting for 42.7% of the interchange accidents. It shows that the number of accidents in SSIs is significant, and their safety deserves attention.

Then, the characteristics of accident types in the SSI areas are explored. This paper divides the kinds of accidents into eight categories: Rollover, rear-end collision, scrapping, collision, collision with fixed objects, pedestrian collision, spontaneous combustion, and others. In addition, a control group was set up to intuitively reflect the characteristics of accident types in the SSI area. In Section I, the area from the No.4 interchange to the No.5 interchange was set as the NSI area. Similarly, the area from No.6 to No. 7 interchange in Section II was another NSI area. The statistical results are shown in Figure 6.

Figure 6a shows that rear-end collisions are the main accident form in the interchange section of expressways, and the number of such accidents is far more than other types of accidents. This result verifies the conclusions in the literature [4]. Furthermore, the number of rear-end collisions in the SSI is higher than in NSI. Due to the shortened spacing distance, drivers have specific cognitive difficulties, and the degree of vehicle interaction is even more severe. Entering vehicles, exiting vehicles, and straight vehicles need to complete driving tasks within a very short distance, which is bound to affect the speed of cars, resulting in speed differences and increasing the probability of rear-end accidents.

On the other hand, the number of collisions with fixed objects and rollover accidents in the SSI is also higher than in the NSI. In the SSI, the driver has specific operational difficulties. To complete the driving task, the driver has to engage in risky driving behaviors, such as swerving and braking sharply, which may increase the probability of collision with fixed objects. Once the vehicle is out of balance, a rollover may occur. At the same time, the vehicle’s driver in the rear will likely take coping behaviors such as slamming the direction, which will make the car more likely to cause accidents such as rear-end collisions, rollovers, and collisions with fixed objects. Compared with NSI, SSI should focus more on preventing rear-end collision, rollover, and collision with fixed objects. By adding signs and markings, the vehicle should be guided to drive on the SSI at a reasonable speed and complete the corresponding driving operations.

Subsequently, the difference in the number of vehicles involved in accidents is analyzed. The statistical results are shown in Figure 7. Because there is not much difference in the traffic volume, it is reasonable to compare and analyze the number of cars in the same section. Similar to previous studies, multi-vehicle accidents account for much of the total number of accidents, especially two-vehicle accidents. In addition, the number of two-vehicle accidents in the SSI is more than that in the NSI, indicating that the degree of interaction between vehicles is higher. At the same time, the above viewpoints verified that rear-end collisions often occur in the SSI.

Finally, the difference between serious accidents and non-serious accidents is analyzed. According to existing studies [33,34], this paper considers fatal accidents and accidents with severe injuries as serious accidents, and minor injury accidents and property loss accidents are viewed as non-serious accidents. The statistical results are shown in Figure 8. In Section I, regardless of serious or non-serious accidents, the accident frequency in the SSI is higher than in the NSI. In Section II, non-serious accidents in the SSI were more severe than in the NSI. Under similar traffic conditions, SSI is more prone to non-serious accidents, while NSI is more prone to serious accidents. However, SSI and NSI are identical in the number of serious accidents. Thus, it is necessary to take some measures in the SSI to prevent serious accidents.

In general, SSI and NSI have some similarities and differences. They all belong to the interchange, so there is a certain similarity in the type of accident. For example, rear-end collisions are the main types of accidents, and multi-vehicle accidents are the main accidents. Simultaneously, there are some differences between the two interchange types. For example, more collisions with fixed objects occurred in the SSI, and two-vehicle accidents are the primary accidents. The above findings can provide some support for traffic safety management.

### 5.3. Accident Severity Casual Analysis in SSI Area

This section will rank the importance of the accident severity influencing factors. Moreover, the difference between the different interchange types will be obtained, providing a specific basis for improving traffic accident prevention.

#### 5.3.1. Variable Selection

It is necessary to choose variables according to the actual needs and conditions. The variables should be independent and describe the causes of traffic accidents as comprehensively as possible. Compared with the indicators in Table 2, the variables selected in this section are micro-level data. The selected variables are used to describe the exact situation of each traffic accident. According to the existing studies, 12 variables are considered, as shown in Table 3. The explanation can be found in Appendix A. Every variable is a discrete variable except TIC. TIC is a continuous variable whose value depends on how long it takes to deal with a traffic accident. The values of discrete variables have been given in Table 3, and the number before each category is the value of that category.

The variance inflation factor (VIF) is used to test the 12 variables to ensure that the variables are not affected by multicollinearity. They strongly correlate if the correlation coefficient exceeds 10 [34]. As shown in Table 4, the correlation coefficients of all variables are less than 10, so all variables should be retained.

#### 5.3.2. Model Result

In this section, the accident data are randomly divided into two groups, a training data set and a test data set, with a specific ratio (70/30 in this study). After data preprocessing, 229 SSI accident data and 278 NSI accident data were adopted from Sections I and II. According to accident data, LSSVM, RF, and LR models have been developed with the help of SPSS 18.0 software (International Business Machines Corporation, New York, NY, USA) for two interchange areas, and the related information is shown in Table 5. Then, the contribution of different factors is explored in the two regions regarding the accident severity. Finally, the results of the three models are compared and analyzed to determine the characteristics of accident severity. On the other hand, both AUC and Gini are used to describe the classification power of the models. AUC is the area under the receiver operating characteristic (ROC) curve enclosed by the coordinate axes. The closer the AUC value is to 1, the stronger the classification power of the model. The Gini refers to the probability that two samples belong to different classes. The model has strong explanatory power when the Gini value exceeds 0.6. The conversion relationship between the two is Gini = 2 × AUC − 1.

First, the LSSVM model was constructed with 12 variables of SSI. As shown in Table 5, the accuracy of the train set, test set, and validation set was 89.4%,89.3%, and 92.1%, respectively. In addition, the AUC value and Gini coefficient are relatively high, indicating that the model has a satisfactory fitting degree. Similarly, the LSSVM model is also applicable to NSI variables. The order of importance of accident factors is shown in Figure 9. In the SSI, TIC, TCT, NOV, SCO, and RFC are the top five factors. In the NSI, the top five factors are TIC, TCT, TAT, MON, and ROA. TIC and TCT are the common influencing factors of the two interchange areas. The priority of TIC is much higher than other factors, reflecting the fact that serious accidents need a longer time to handle. There may be casualties or severe property losses when a serious accident occurs. Therefore, it takes a lot of time for police, ambulance, and road maintenance personnel to deal with it simultaneously. Non-serious accidents, such as scrape accidents with no casualties or significant property damage, can be handled quickly. Moreover, TCT is one of the critical factors affecting the severity of accidents. Previous studies have shown that large vehicle accidents are mostly severe due to their large mass and weak braking performance [15]. Thus, large vehicles have a particularly negative impact on traffic safety.

Due to the limited spacing, incoming and outgoing vehicles have frequent weaving behavior. When safety space is not guaranteed, the car is prone to traffic accidents; the more vehicles involved in a traffic accident, the more likely the number of occupied lanes. If the above situation occurs, the accident will become more serious. In addition, the road surface friction coefficient will have a positive effect on serious accidents. If the friction coefficient of road surface decreases, it may interfere with the driver’s judgment and affect car operation. Therefore, TIC, TCT, NOV, SCO, and RFC are seen as the main factors. In the NSI, the impact of NOV and SCO on accident severity in NSI areas is not particularly significant. The TAT, MON, and ROA have a particular influence on the accident’s severity. It is mutually corroborated with existing research [16]. 

As shown in Table 5, the RF model’s accuracy and fitting degree are satisfactory. The importance ranking of accident factors is shown in Figure 10. According to the analysis based on the decision rules for serious accidents, serious accidents will likely occur in the SSI area from February to December when the TIC is more than 64 min. In the NSI area, TIC > 60 min is prone to serious accidents. Contrary to LSSVM’s results, the MON is another main factor. In addition, SCO influences the severity of the accident in the SSI, and TAT is the influencing factor in NSI areas. The above results are similar to the LSSVM model results.

Finally, the LR model is established. Its evaluation indicators are satisfactory, and the model’s accuracy is higher than LSSVM and RF. The order of essential factors is shown in Figure 11. The top five influencing factors in the SSI are TIC, TAT, TCT, RFC, and NOV. The top five influencing factors for the NSI are TIC, TCT, TAT, NOV, and SCO. By comparison, it is found that although the results given by the model include TIC, TAT, TCT, and NOV, there are differences in the ranking of some indicators. In addition, the RFC factor is more significant in the SSI. If the road is wet, it will increase the operation difficulty of the driver and even cause roll-away, side-sliding, and multi-vehicle collision accidents. In the NSI, SCO is significantly similar to the results given by the RF model. When multi-vehicle accidents occur, they are most likely to be severe and occupy multiple lanes easily.

In general, there are some similarities in the causes of accidents in the two kinds of interchange areas. Firstly, TIC is the common influencing factor of severe accidents, so the efficiency of accident handling should be improved as much as possible to prevent serious accidents. Another common influencing factor is TCT, verified mutually with existing studies. Large vehicles will affect the safety of the interchanges, so they should be strictly regulated [15]. On the other hand, there are some differences between the two interchange types. The RFC and NOV are significant factors requiring extra attention in the SSI. Some measures should be taken to prevent serious accidents when the road is slippery. Simultaneously, it is necessary to set up signs and lines to remind vehicles to maintain reasonable spacing to avoid multi-vehicle accidents. In the NSI area, the TAT is relatively significant, verifying the scientific nature of existing studies and providing a theoretical basis for accident prevention.

## 6. Discussion

This research systematically analyzes the characteristics of traffic accidents in the SSI. Based on the analysis results, the results provide some implications for the interchange accidents.

From a macroscopic point of view, this paper points out that frequent accidents occur in expressway interchanges. Consistent with current research results [31,32], DEN and AADT are the leading causes of expressway accidents, and rear-end collision is the main form of freeway accidents. In addition, MSD has a specific impact on accidents. The accident statistics show that the SSI accident volume and *NAP* values are higher than NSI. Therefore, the safety of SSI deserves the attention of traffic managers and researchers. From the perspective of accident severity, the top five critical variables in SSI are TIC, TAT, TCT, RFC, and NOV. Therefore, the negative impact of the above factors should be considered in traffic design and traffic safety management. In addition, it is mentioned in reference [1] that the accident risk is relatively highest in three-quarters of the outside lane of the SSI. Therefore, when designing SSI, we should guide different types of vehicles by setting up signs before entering the SSI. Warning signs should remind drivers to maintain a safe distance and speed. The color road surface of the outermost lane can increase the friction coefficient of the road surface so that it can have a positive effect on traffic safety prevention. 

The classification accuracy of most models is higher than 87%, which verifies the effectiveness of these models. It may be because traffic accident data provides rich information, which can give more details for accident classification. LSSVM is suitable for classifying data with a small sample size, and it can help us grasp critical samples and “eliminate” a large number of redundant samples. Moreover, this method is destined to be simple in algorithm and has good “robustness”. The RF model results reveal the rules with the top-ranking parameters. Furthermore, it is more suitable for exploring key variables with fewer influencing factors.

The LR model is parametric, and its accuracy is better than that of LSSVM and RF in this classification task. There is some difference between this result and existing conclusions [23]. Firstly, the prediction accuracy of the parameter model is satisfactory, which avoids the problem of over-fitting the model and has a good performance in the accident risk assessment. Secondly, the parameter model is suitable for capturing the internal relationship between multiple explanatory variables and accidents. The model estimation results are intuitive, functional expressions that can accurately provide the probability of accidents and estimate each dependent variable’s marginal effect on accidents. It is more suitable for practical engineering applications. On the other hand, although the accident data are rich, the volume of information is not large, so it is more suitable to adopt the parameter model. 

## 7. Conclusions

To explore the characteristics and cause of the SSI accident, this paper used *NAP* as an evaluation index to evaluate the safety of the four sections in the expressway. Secondly, eight indicators, such as MSD, were selected to explain the significant factors in expressway traffic safety with the help of the LSSVM model. Thirdly, the accident data of the SSI and the NSI were analyzed to clarify the differences in accident characteristics. Finally, the LSSVM, RF, and LR models were developed to study the causes of serious accidents in the SSI and NSI. It can be concluded that:(1)There is a particularly negative impact on the safety of the expressway in the SSI area. The value of inner ring *NAP* in Sections I and II containing SSI is 27.2 and 33.7, respectively, higher than in other sections. Therefore, some safety measures should be taken to improve traffic safety in the SSI area.(2)The MSD has a specific negative impact on expressway safety. This paper ranked the importance of the eight indicators, such as MSD and AADT, by constructing the LSSVM model. The importance of DEN, AADT, and MSD was significant. Therefore, paying attention to the MSD on expressway safety is necessary.(3)TIC is the significant index of serious accidents in the SSI and NSI. This paper constructs the LSSVM, RF, and LR models, respectively. The importance of the TIC index is far greater than other indexes. Therefore, it is necessary to improve the efficiency of accident handling to reduce the severity of accidents.(4)RFC and NOV are relatively important indices of serious accidents in the SSI. LSSVM and LR models reflect that the RFC and NOV are relatively substantial, so extra attention should be paid to the SSI area’s road condition and vehicle spacing distance.

However, there are still some limitations to this research. In the future, the study will continue in the following directions to further improve the relevant research:(1)In future studies, the amount and type of data will be enriched, and the relationship between drivers’ factors and accidents will be fully considered.(2)Future studies will quantitatively analyze the impact of interchange spacing distance on traffic safety and find the safety threshold.

## Figures and Tables

**Figure 1 ijerph-19-09938-f001:**
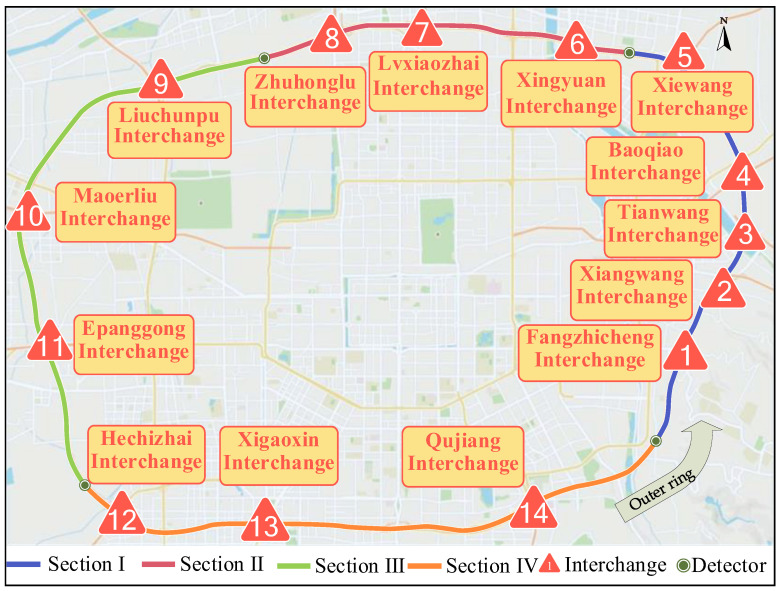
Schematic diagram of G3001 expressway.

**Figure 2 ijerph-19-09938-f002:**
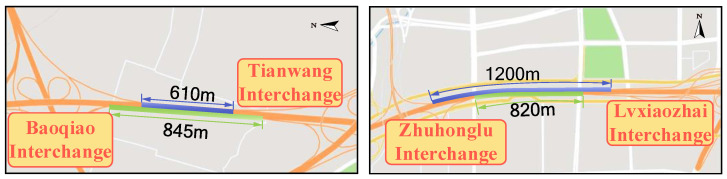
Schematic diagram of SSI.

**Figure 3 ijerph-19-09938-f003:**
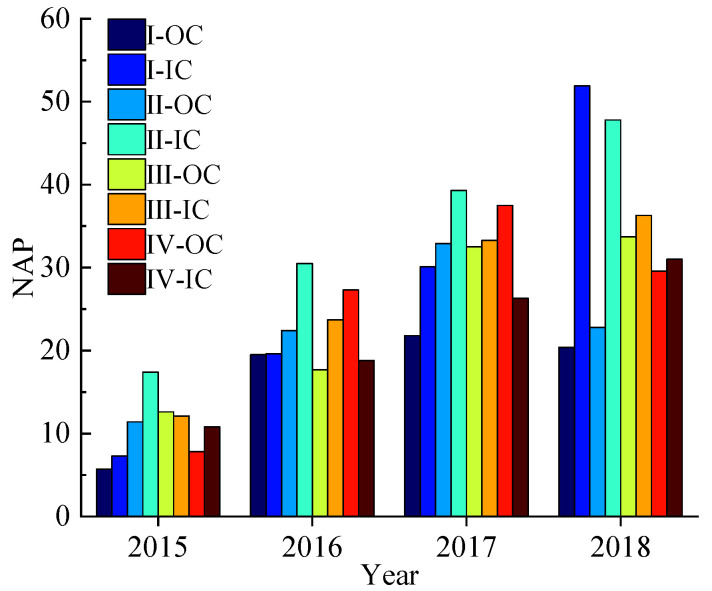
*NAP* value distribution.

**Figure 4 ijerph-19-09938-f004:**
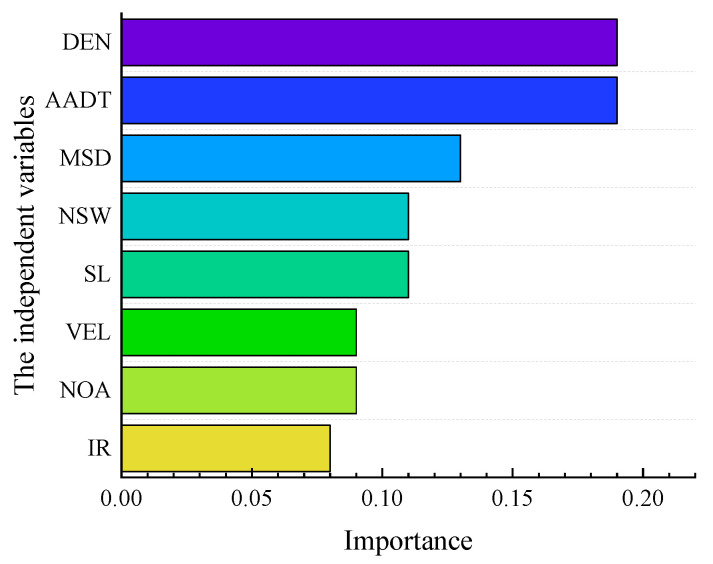
Order of importance of influencing factors.

**Figure 5 ijerph-19-09938-f005:**
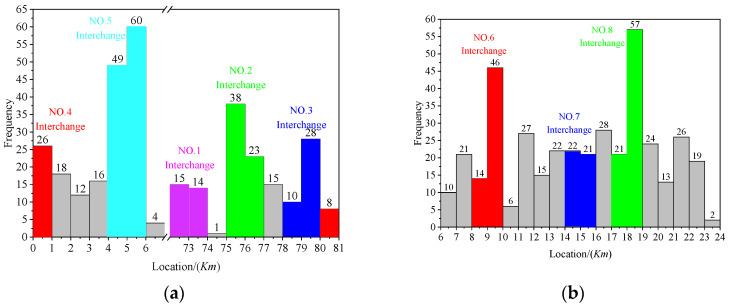
(**a**) Accident mileage distribution in Section I; (**b**) Accident mileage distribution in Section II.

**Figure 6 ijerph-19-09938-f006:**
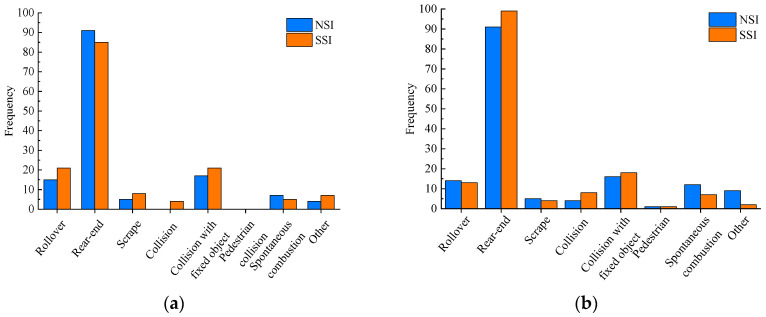
(**a**) Accident type in Section I; (**b**) Accident type in Section II.

**Figure 7 ijerph-19-09938-f007:**
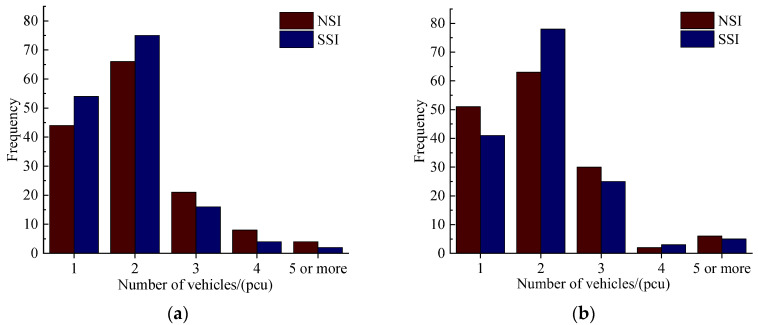
(**a**) Number of vehicles in Section I; (**b**) Number of vehicles in Section II.

**Figure 8 ijerph-19-09938-f008:**
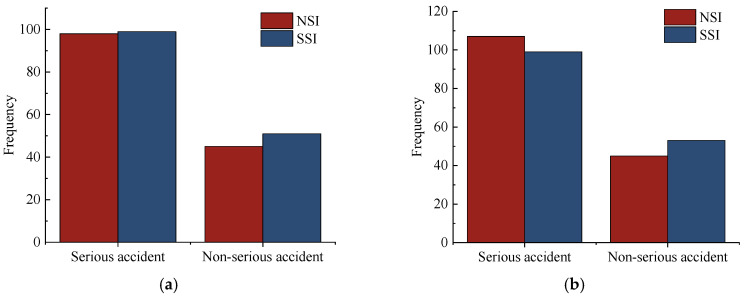
(**a**) Number of accident severity in Section I; (**b**) Number of accident severity in Section II.

**Figure 9 ijerph-19-09938-f009:**
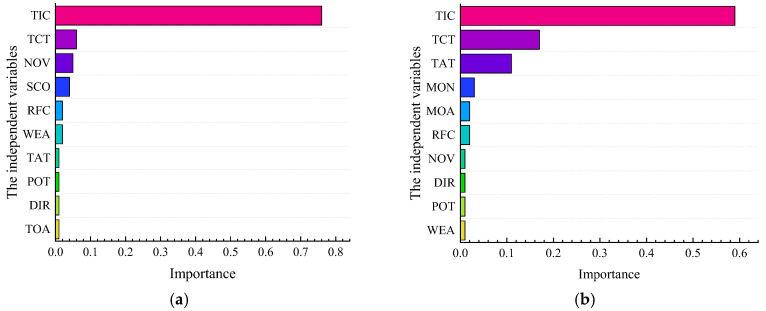
(**a**) The result of the LSSVM model in the SSI area; (**b**) The result of the LSSVM model in the NSI area.

**Figure 10 ijerph-19-09938-f010:**
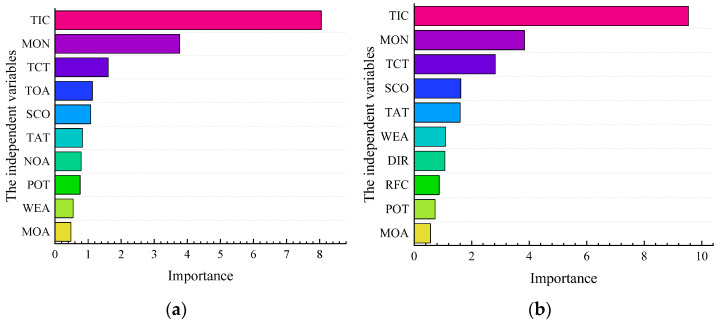
(**a**) The result of the RF model in the SSI area; (**b**) The result of the RF model in the NSI area.

**Figure 11 ijerph-19-09938-f011:**
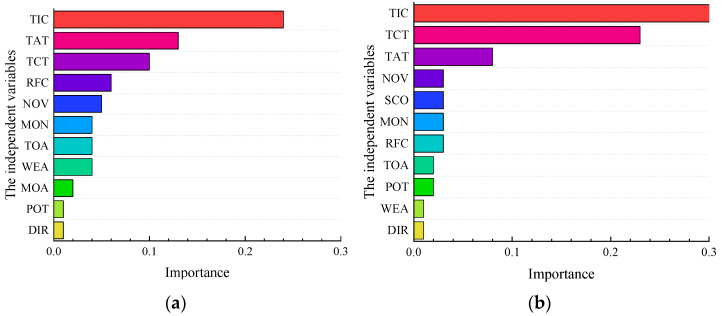
(**a**) The result of the LR in the SSI area; (**b**) The result of the LR in the NSI area.

**Table 1 ijerph-19-09938-t001:** Details of section division.

Section	Length	Interchange Information
I	17 km	1, 2, 3, 4, 5
II	18 km	6, 7, 8
III	22.3 km	9, 10, 11
IV	24.5 km	12, 13, 14, 15

**Table 2 ijerph-19-09938-t002:** The values of the independent variable and dependent variable.

Location	Year	*NAP*	AADT (pcu/d)	VEL (km/h)	DEN (pcu/km)	IR (%)	MSD (km)	SL (km)	NSW (%)	NOA (Count)
I-OC	2015	5.7	1.55	72.7	14.5	24	1.98	14.3	45	11
	2016	19.5	1.90	75.2	14.9	24	1.98	14.3	37	43
	2017	21.8	2.54	78.3	16.5	24	1.98	14.3	32	65
	2018	20.4	2.75	80.4	16.3	24	1.98	14.3	31	54
I-IC	2015	7.3	1.45	65.9	7.8	31	2.09	14.3	29	17
	2016	19.6	1.81	72.1	11.2	31	2.09	14.3	26	34
	2017	30.1	2.62	76.9	17.9	31	2.09	14.3	16	50
	2018	51.9	3.32	83.9	19.6	31	2.09	14.3	22	63
II-OC	2015	11.4	2.44	74.2	15.4	30	3.1	16.77	45	20
	2016	22.4	2.96	74.3	19.2	30	3.1	16.77	37	46
	2017	32.9	3.35	75.4	20.5	30	3.1	16.77	18	73
	2018	22.8	3.93	74.9	25.5	30	3.1	16.77	17	54
II-IC	2015	17.4	1.92	75.5	14.7	17	2.91	16.77	32	22
	2016	30.5	2.37	74.9	14.5	17	2.91	16.77	26	43
	2017	39.3	2.35	76.2	14.3	17	2.91	16.77	18	60
	2018	47.8	3.11	74.6	20.6	17	2.91	16.77	25	76
III-OC	2015	12.6	4.6	82.8	16	13	5.15	23	33	21
	2016	17.7	3.38	75.8	11.9	13	5.15	23	23	22
	2017	32.5	1.19	76.8	8.8	13	5.15	23	28	46
	2018	33.7	0.45	65.9	4.2	13	5.15	23	26	51
III-IC	2015	12.1	5.17	82.6	16	13	5.45	23	13	15
	2016	23.7	3.78	77.1	13	13	5.45	23	17	29
	2017	33.3	1.71	77.5	13	13	5.45	23	21	48
	2018	36.3	0.67	69.1	8.4	13	5.45	23	18	39
IV-OC	2015	7.8	0.75	66.1	18.4	16	2.28	25.7	19	21
	2016	27.3	1.95	66.6	24	16	2.28	25.7	27	44
	2017	37.5	2.98	69.8	21.6	16	2.28	25.7	18	83
	2018	29.6	4.27	68.8	31.2	16	2.28	25.7	23	106
IV-IC	2015	10.8	0.28	76.7	14.3	18	2.82	25.7	33	27
	2016	18.8	1.92	73.1	23.5	18	2.82	25.7	34	62
	2017	26.3	3.44	71.9	24.2	18	2.82	25.7	15	92
	2018	31	5.15	67.2	37.7	18	2.82	25.7	20	111

**Table 3 ijerph-19-09938-t003:** List of accident influence factors.

Variable	Values of Categories	Variable	Values of Categories	Variable	Values of Categories
ROA	1. Line	POT	1.day	TOA	1. Working days
	2. Bend		2.night		2. Day off
MON	1. January 2. February… 12. December	RFC	1. Normal 2. Road surface slippery 3. Ice on the road	DIR	1. inner circle 2. outer circle
WEA	1. Sunny 2. Cloudy 3. Snow 4. Rain 5. Fog	NOV	1. One 2. Two 3. Three 4. Four 5. More than five	TIC	Actual time/minute
TCT	1. Passenger car	TAT	1. Rollover accident 2. Rear-end accident 3. Scrape accident 4. Collision 5. Collision with fixed object accident 6. Pedestrian collision 7.Spontaneous combustion 8. Other accidents	SCO	1. Lane one 2. Lane two 3. Lane three 4. Emergency lane 5. All 6. Emergency lane + lane one 7. Emergency lane + lane two 8. None
	2. Bus				
	3. Van				
	4. Large truck				
	5. Semi-trailer				
	6. Minivan				
	7. Special vehicle				
	8. Passenger car-bus mix				
	9. Van-Large truck mix				
	10. Passenger-truck hybrid				
	11. Multiple models				

MON: Month; WEA: Weather; TCT: The car type; POT: Period of time; RFC: Road form condition; NOV: Number of vehicles; TAT: The accident types; TOA: Time of accident; DIR: Direction; ROA: Road alignment; TIC: Time consuming; SCO: Scope.

**Table 4 ijerph-19-09938-t004:** The VIF value of the variables.

Variables	VIF	Variables	VIF	
MON	4.576427	WEA	3.549679	
TOA	8.675889	TCT	2.672981	
POT	8.274022	NOV	5.506071	
TIC	1.327032	TAT	3.429347	
ROA	8.377142	SCO	3.153264	
DIR	8.85624			

**Table 5 ijerph-19-09938-t005:** Parameters related to the three models.

Section		SSI			NSI		
Three models for severity		LSSVM	RF	Logistic	LSSVM	RF	Logistic
The number of predictive variables entered		12	12	12	12	12	12
Number of predictive variables in the final model		12	12	12	12	12	12
Normalized type		L2	-	-	L2	-	-
Penalty parameter (Lambda)		0.1	-	-	0.1	-	-
Train set	Accuracy	89.4%	90.9%	97.8%	97%	94%	98.1%
	AUC	0.997	0.997	0.995	0.998	0.999	0.992
	Gini	0.994	0.994	0.988	0.996	0.998	0.984
Test set	Accuracy	89.3%	88%	98.8%	87.8%	92.8%	98.7%
	AUC	0.988	0.958	0.873	0.876	0.972	0.745
	Gini	0.996	0.916	0.746	0.752	0.944	0.49
Validation set	Accuracy	92.1%	86.7%	96.4%	88.1%	92.4%	98%
	AUC	0.959	0.995	0.825	0.944	0.987	0.814
	Gini	0.918	0.99	0.649	0.887	0.974	0.628

## Data Availability

The data presented in this study are available on request from the corresponding author. The data are not publicly available due to privacy.

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
