# Peer review of "Analysis on Risk Characteristics of Traffic Accidents in Small-Spacing Expressway Interchange"

_ijerph, 2022, doi:10.3390/ijerph19169938_

Round 1
Reviewer 1 Report
This is a very interesting and well written paper about risk characteristics of traffic accidents in small-spacing expressway interchange.
I have some comments and suggestions for the authors:
Equation 1: q..L (what is the symbol between q and L)?
You can put Table 3 (if is possible) in the same page in order to be better readable (for example, moving Table 3 in line 166 and then starting the paragraph).
Figure 4a. Yellow colour is very bright and not readable.
Figure 5. The font size for x-axis is too small.
In this paper are used many abbreviations. Could you write a Table as a reference point for all abbreviations?
I agree with the authors that future research in this area is necessary.
Reviewer 2 Report
Summary:
The research attempts to address the increase of accidents in small space interchanges (SSI) by finding several key accident characteristics. In the context, G3001 expressway was chosen as the area of study. Metrics like number of accidents per unit distance of 100 million vehicles (NAP) and mean spacing distance (MSD) were used in the analysis. Subsequently, factors affecting safety in various sections of highway are analyzed using statistical methods, and classification methods such as LSSVM, RF, logistic, etc.
Found the study and the evaluation methods very interesting, and the work presented can be good resource for the researchers in the area.
General comments:
a) Need lot of improvement on the flow of narration, sentence construction, and grammar. See some of the examples given in the section specific comments.
b) State the goal of the research in the introduction section clearly in few sentences or bullet points.
Specific comments:
1. Title:
a) Correct “Small-spacing” to Small-Spacing.
2. Abstract:
a) It is important to provide small description or cue before introducing any topic or abbreviation. For instance, where is G3001 expressway located (city/country)? And, present full form of AADT.
3. Introduction:
a) Lines 30-31: In the phrase, “to meet the city's rapid development…”, it is not clear which city it is in reference; please introduce the city before the sentence.
b) What is the novelty of the research compared to other works in the area? This should be presented in the introduction section.
4. Literature Review:
a) Line 72: Expand IHSDM
b) Line 74-75: Please rewrite the sentence, it is a bit confusing: “The accident statistics method is mainly based on detailed statistical data, and the conclusion is the most objective and reliable after long-term revision.”
c) Line 99: “KIM et al.” should be like “Kim et al.”
d) Expand the short forms CART, SVM, RF, etc. before introduction.
5. Study Area:
a) Dataset: The data collection/preparation, the size of dataset, splitting, training and testing strategies were not presented. In fact, it is the important aspect of the paper which should not be omitted.
6. Methodology (4.3.2):
a) Organize the numbering of LSSVM and RF methods into sections i) and ii) rather than 1) and 2); since these numbers are overlapping with equation numbers (making it a bit confusing).
b) Why the LSSVM, RF, and Logistic methods are chosen (over any other methods available) should be explained more clearly; add more reasons. Also, several equations are presented to explain how the three methods work, but the methods should be explained in text in broader sense (in few steps, how they work on data, output, etc.), before presenting the equations. Also, what kind of algorithms are built to implement these methods should be explained stepwise.
7. Results:
a) Table 5: The “100%” Classification accuracy achieved for logistic regression looks uncommon and is seldom attainable in classification problems. In such instances where 100% is achieved, there is always an underlying problem present, either with the model overfitting, or with datasets; like splitting, sampling, etc. There is always certain amount of error involved in any kind of modelling (SVM, RF, logistic, etc.). In the context, explain how you have achieved 100% accuracy, how big is your dataset, and what is the data split between training and test sets?
b) There is not much presented on how the results are validated for various models. When it comes to accuracy of the model, “validation” is very important, and should be included in results section.
Reviewer 3 Report
Thanks for the opportunity to review this manuscript.
This work intends to investigate the characteristics of traffic accidents in SSI. However, I don’t think that this goal was achieved and I have several drawbacks to point out.
- Why it is important to present some sections data by direction (inner and outer ring, table 2)? It seems that there is a difference (is it significant?) between inner and outer ring. You do not point out this difference.
- According to the definition of NAP, higher the value of NAP worst the safety of the section. Therefore, why do you say that section I have poor safety than the other sections (line 163)?
- Why descriptive statistics is made only for section I and II, and not for all sections if you intend to compare SSI with NSI?
- No statistical tests are used to evaluate the significance of some of the differences. Some of the reported differences seem to be too small to be highlighted as they are in the text.
- None of the used methods is knew. There are several works presenting and using his methodologies. Therefore, this section can be shortened with a reference to some literature. However, more explanation is needed about the dependent variable and about how these methods were used to achieve the main goal of the article (i.e., characteristics of traffic accidents in SSI). What are the considered categories for accident severity and their frequency? What was the training set?
- What data is considered in SSI model? Section I and II or interchange 3, 4, 7 and 8? Same question for NSI model.
- Some variables presented in table 2 have too many categories. Have you merge categories to fit the models? If not, I believe that several categories will have observations. What are the consequences in these models? Are the robust to the lack of observations?
- You miss important measures to compare the models.
Some other comments:
1. The structure of the work needs to be revised. It is confusing and does not follow the usual structure of articles.
2. The relevance of several works cited for this study is not always perceived.
3. There are too much figures and tables.
4. It is not usual to introduce acronyms in the abstract. These must be introduced in the main body of the text. In the template of articles from MDPI journals, there is a subsection where the acronyms that are used throughout the text must be listed, as well as their meaning.
5. In what country is located the G3001 expressway? And is located in a city, or around a city?
6. Some figures and tables are not cited in the text.
7. Check the denominator of formula (1). I advise that you consult some methodological works to see how math is written. For instance, in the end of formula (1) you should use a comma; line 158 should start without capital letter and no indent should be used; before q instead of a dot it should be a comma; before L instead of a dot it should be a comma and the word and.
8. Lines 167-169:
- how can we see in table 3 that spacing distance of multiple interchanges is lower than the standard specification value?
- NAP is relatively high compared to what? Or based on what criterium?
- Why spacing distance of interchanges are likely to affect the number of accidents? How we can see this in table 3?
9. Table 3: the numbers inside the table represent what? Are totals, means, proportions, …? It should be clarified in the label or in the text.
10. Lines 173-176 and Figure 3: what is the dependent variable(s)?
11. Line 180: I’m not sure “causing” is the right word. I don’t think that you can say that they are the cause...
12. Figure 4: If section I has 17,8 km and section II 16 km (according to table 1), how can the values in the x axis? (for instance, in Figure 4b you have 18 km instead if 16 km. There is a much high frequency in kms 4-6 (Figure 4a), and none is said about that.
13. There are two table 3. In the second table 3, the titles of the second table, instead of values title, it should be values of categories.
14. Figure 8: you used the Spearman coefficient. This coefficient can only be used if your data is at least ordinal, which is not the case…
15. There is no need to repeat the reference to SPSS program over the text. One time is enough.
16. English: major revisions. I’m not english native, but I found several confused sentences, incomplete sentences, unusual words.
Reviewer 4 Report
Discussion could be a bit more focused on how to improve the overall highway road safety by design, given the findings.
Page 9, Line 275. Table 3 instead of Table 4?
Figure 8. How the values of the aforementioned variables can be interpreted if those correspond to categorial (nominal) type variables?
Line 379
TIC and TCT appear as the critical factors affecting the severity of accidents, but their actual interpretation remains unclear when looking at the definitions from Table 3.
Reviewer 5 Report
1. Post a histogram under the Table 2 will be more visible to show the accidents distribution at different section during investigation years.
2. The name of horizontal ordinate is needed for Figure 6.
3. How to define the serious accidents and non-serious accident need a further explanation under Figure 7.
4. The name of Table 2 is same with Table 3, it is suggested that change the name of Table 2 to the parameter values of accident influence factors. Besides, an explanation of the relationship or difference between the 8 factors in Table 2 and the 12 factors in Table 3 is needed.
5. References for the methodology part such as the LSSVM, Random Forest, Logistic model should be marked in your text.
Round 2
Reviewer 2 Report
Thank you for revising. The paper looks improved than before, however, there should be improvements in terms of emphasizing and explaining the key results and performance indicators.
1. For instance, the SPSS results (Table 5) should be explained clearly rather than simply stating the end result. For example, AUC measure (expand this term before introducing) and Gini coefficient are presented, however, they were not introduced or neither explained about their significance. AUC is an important statistical measure to compare the predictive power of two or more different statistical classification models when used over the same data set or sample.
2. Likewise, what is the significance of Gini and its relation to AUC. What are the Gini and AUC value ranges that indicate better classification/predictive performance? These should be stated briefly before stating “model has a satisfactory fitting degree,” so that readers can understand why.
3. Line 555: Please use “limitations” in place of “defects” in the sentence, “However, there are still some defects in this research.”
Reviewer 3 Report
I am grateful for the answers given by the authors that allowed clarifying some stages of this study. The revised version has several improvements, both in presentation and English, but there are still several improvements to be made.
The need to revise the English language remains, as there are still several sentences that are unclear or have inadequate words. Below I will point some, but there are more.
In the previous revision I pointed that you could not use Spearman correlation with that kind of variables. The variables in table 4 are categorical, most of them nominal. Therefore, you can not use Spearman’s correlation neither Pearson correlation (Figure 9). You have to use other type of coeficients! However, if your idea is to check for multicollinearity, why you have not used the GVIF?
I’m concerned about the choices you made made when you used the variables presented in table 4. In the text you write that the variables are discrete and that TIC was discretized. However, as I said before, these variables are qualitative not quantitative. The results form LSSVM, RF e LR may not be the same if you consider the variables as quantitative instead of qualitative.
The structure does not follow the requirements of the journal. According to the IJERPH Instructions for Authors “all manuscripts must contain the required sections: […], Abstract, Keywords, Introduction, Materials & Methods, Results, Conclusions, …”. Your article does not have the Materials & Methods neither Results section. They are a subsection of a major section.
Detailed coments:
ABSTRACT:
As I referred in the previous revision, in the abstract you should define just the abbreviations that you need for the abstract, e.g., SSI, MSD and LSSVM. All the others should be presented in the text, as well as the abbreviations used in the abstract, in the first time that they appear.
INTRODUCTION:
- Line 43: please review the sentence because “are developed to the relationship” does not make sense.
- Line 53: what do you mean by “statistical characteristics”? Please clarify.
- Line 61-63: Review the sentence. The way it it written, it seems more a conclusion/discussion than an explanation of what you will do.
LITERATURE REVIEW:
- Line 78-79: If Qi used IHSDM software as it is refered in line 76-77, you can’t write that this is a statistical method. Or IHSDM is a statistical method or it is a software. As far as I know, IHSDM is a software tool and there are statistical models embedded in IHSDM. Therefore, please rephrase this sentence.
- line 81-82: I think this is the first time you present LR; hence it should be “logistic regression (LR)”. I think that instead of “accident statistics data” it should be “accidents data”.
- There are several works that does not seem to be relevant to the analyze the risk characteristics of accidents in SSI. For instance: what is the relevance of the works [14], [16], …? You should focus on works about expressway traffic accidents.
STUDY AREA:
- Lines 141-142: please, rephrase the sentence.
- Figure 2: change the order of the figures, i.e., in the left should be interchanges 3 and 4, and in the right interchanges 7 and 8, to be in accordance with the order in the text.
RISK FACTORS SYSTEMATIC ANALYSIS IN SSI
- Lines 154-155: What previous studies? Add references, please.
- Figure 3 and table 2 represent the same information. Keep just one of them.
- Line 173: It is not clear what you want to say with the sentence “Similarly, … SSI area”.
- Lines 170-175: in this paragraph you should also refer that “there are differences in traffic flow data in different direction but there is no great difference in road geometry and traffic environment between inner and outer circles”, as you wrote in the response 1 of the review.
- Lines 183-185: if NAP is the dependent variable (line 184), you are not analyzing the cause of accidents (line 183). You are analyzing the accident rate or the safety of the section.
- Table 3: correct the label. If NAP is the dependent variable it is not an influence factor.
- Lines 193-194: remove the sentence “DEN … state” since it repeats the first sentence.
- Line 235: check if it is Figure 5.
- Line 294: please correct. A discrete variable is a quantitative variable. Categorical variables are qualitative variables. You can transform a discrete variable in a qualitative variable, but not the contrary.
- Figure 9: use an appropriate correlation coefficient.
- Line 309: please, reinforce that they are only considering 2 categories for severity (serious vs. non-serious)
- Lines 381-384: unclear sentences. Review, please.
- Line 392: Shouldn’t the references [23, 31, 33] be at the end of the sentence?
- Lines 411-412: review mathematical writing
- Line 416: 0.5 for every LR model. This is not true. The boundary is chosen according to some criteria, and can change between adjusted models. Please improve the sentence.
- Table 5: weren’t all models adjusted for severity? If yes, remove line “the target field” and in the label of table 5 add “… three models for severity”
Appendix A: Switch the order of the columns, that is, first present the abbreviation and then its meaning. For ease of reading, acronyms should be presented in alphabetical order.
Other observations
- I think that instead of “scholars” is should be “researchers” (line 67, 81, and many others)
- Line 73: instead of “Wang et al. [11] demonstrated” it should be “Wang et al. [11] used”
- Line 137: instead of “Besides, 1597” it should be “A total of 1597”
- Line 165: instead of “owns two” it should be “has two”
- Line 218: instead of “most among” it should be “highest among”
